# Novel Exosomal miRNA Expression in Irradiated Human Keratinocytes

**DOI:** 10.3390/ijms252212477

**Published:** 2024-11-20

**Authors:** Hebah Almujally, Nizar Abuharfeil, Aseel Sharaireh

**Affiliations:** 1Department of Biotechnology and Genetic Engineering, Jordan University of Science and Technology, Irbid 22110, Jordan; haalmujally20@sci.just.edu.jo (H.A.); harfeil@just.edu.jo (N.A.); 2Department of Restorative Dentistry, School of Dentistry, The University of Jordan, Amman 11942, Jordan

**Keywords:** skin aging, sun solar simulator irradiation, oxidative stress, ROS, keratinocyte-derived exosomes, exosomal miRNAs

## Abstract

The epidermis, the outer layer of the skin, relies on a delicate balance of cell growth and keratinocyte differentiation for its function and renewal. Recent research has shed light on exosomes’ role in facilitating skin communication by transferring molecules like miRNAs, which regulate gene expression post-transcriptionally. Additionally, these factors lead to skin aging through oxidative stress caused by reactive oxygen species (ROS). In this research project, experiments were conducted to study the impact of Sun2000 solar simulator irradiation on exosomal miRNA profiles in HEKa cells. We hypothesized that acute oxidative stress induced by solar simulator irradiation would alter the expression profile of exosomal miRNAs in HEKa cells. The cells were exposed to different durations of irradiation to induce oxidative stress, and the levels of reactive ROS were measured using the CellROX Deep Red flow cytometry assay kit. Exosomes were isolated from both control and irradiated cells, characterized using DLS and SEM techniques, and their miRNAs were extracted and analyzed using qPCR. Solar simulator irradiation led to a time-dependent increase in intracellular ROS and a decrease in cell viability. Exosomal size increased in irradiated cells. Fifty-nine exosomal miRNAs were differentially expressed in irradiated HEKa cells, including hsa-miR-425-5p, hsa-miR-181b-5p, hsa-miR-196b-5p, hsa-miR-376c-3p, and hsa-miR-15a-5p. This study highlights the significant impact of solar radiation on exosomal miRNA expression in keratinocytes, suggesting their potential role in the cellular response to oxidative stress.

## 1. Introduction

Skin is the largest organ in the human body, making up approximately 15% of an adult human’s total body weight, and spans 1.8 m^2^ of surface area. It acts as a protective barrier and also helps maintain homeostasis by preventing the loss of electrolytes, fluid, and proteins, while also assisting in immune response and sensory perception [1]. Skin is comprised of three distinct layers: the epidermis, dermis, and hypodermis, which work together to provide mechanical defense, photoprotection, immunosurveillance, nutrient metabolism, repair, and rejuvenation. The epidermis is the most superficial layer of the skin, and its stratified squamous epithelium consists of keratinocytes (up to 95%) that coordinate the layered structure and continuous regeneration of the epidermis, along with dendritic cells such as melanocytes, Langerhans cells, which are crucial for the immune barrier of the epidermis, and Merkel cells, which constitute the sensory nerve component of the skin [2]. Depending on the morphology and proliferation status of keratinocytes, the epidermis is sub-divided into four distinct layers, characterized by progressive keratinocyte differentiation (from outermost to innermost): stratum corneum, stratum granulosum, stratum spinosum, and stratum basale.

Epidermal keratinocytes undergo a special form of terminal differentiation and programmed cell death known as cornification. It is responsible for transforming proliferative basal keratinocytes into enucleated, flattened corneocytes that form the outermost stratum corneum. Through the asymmetric division of basal keratinocytes, transient amplifying cells detach from the underlying basement membrane, initiating terminal differentiation. This process is accompanied by dynamic shifts and specific expression patterns of marker genes at each differentiation stage under the control of transcription factors such as p63, NF-κB, myc, and Notch [3]. It is a highly regulated form of PCD with distinct features compared to the classical apoptotic pathway observed in other tissues. In contrast to apoptosis, cornification does not involve fragmentation of the dead cell into apoptotic bodies nor does it trigger phagocytosis by immune cells, but rather functions as an essential component of the stratum corneum barrier [4,5]. The balance between self-renewal and differentiation must be regulated by progenitor and stem cells, which depend on numerous internal and external factors [6]. One crucial aspect is the intercellular communication between cells and tissues. Recently, scientists discovered that extracellular vehicles (EVs) released into interstitial spaces play a significant role in mediating communication between keratinocytes and other skin cells. These EVs transfer essential molecules between neighboring cells, influencing important cellular behaviors such as proliferation, migration, and pigmentation [7].

EVs are small membrane-enclosed vesicles released by most cell types. They can be classified into three categories based on their biogenesis: apoptotic bodies, macrovesicles, and exosomes. Exosomes are the smallest, are an abundant sort of EVs secreted into the extracellular space by most cell types, and are found in many body fluids and cell culture supernatants. Their diameters range between 50 and 150 nm; they are products of endocytosis and exocytosis [8,9]. They are composed of a lipid bilayer and contain soluble and membrane-bound proteins, genomic DNA, RNA (such as mRNA, miRNAs, and other small RNAs), lipids, and metabolites derived from the parent cells [10]. A recent discovery in exosome biology unveiled the presence of functional RNA molecules, particularly miRNAs, encapsulated in exosomes with a specific EXO-motif GGAG, which is present in miRNAs and recognized by the heterogeneous nuclear ribonucleoprotein A2B1 protein, thereby controlling their loading to exosomes [11].

MiRNAs are short, single-stranded RNA molecules, typically 18–25 nucleotides long, that play a crucial role in regulating gene expression at the post-transcriptional level. They can interact with mRNA, leading to the degradation of target mRNA or the inhibition of target mRNA translation. Most miRNAs are transcribed from DNA sequences into primary miRNAs and then processed into precursor miRNAs and mature miRNAs. While miRNAs typically interact with the 3′ untranslated region of target mRNAs to suppress expression, interactions with other regions, including the 5′ UTR, coding sequence, and gene promoters, were also reported [12]. Exosomal miRNAs play a crucial role in regulating cellular processes in the epidermis, such as cell proliferation and differentiation. Among the most abundant miRNAs in keratinocyte-derived exosomes are the let-7 family, including hsa-miR-22, hsa-miR-27b, and hsa-miR-21, along with hsa-miR-200, hsa-miRNA-203, and hsa-miRNA-205. Specifically, hsa-miR22 directly suppresses transcription factors involved in keratin gene expression, affecting keratinocyte differentiation. Additionally, let-7b downregulation in human epidermal stem cells interferes with the cell cycle transition, ultimately inhibiting cell proliferation and inducing terminal differentiation [13].

Keratinocyte cornification, a process marked by growth inhibition, altered gene expression, and a shift toward a more mature phenotype ultimately leads to a state of irreversible growth associated with the normal aging process. A guideline of the German Dermatological Society defines skin aging as a continuous, cumulative loss of certain properties present in juvenile skin and accounting for tautness, stretchability, elasticity, and pigmentation influenced by intrinsic and extrinsic factors [14,15]. Intrinsic aging, also known as chronological aging, occurs naturally over time because of accumulated oxidative damage. This damage comes from reactive oxygen species (ROS), which are partially reduced metabolites of molecular oxygen generated as products of metabolic reactions or as by-products of various cellular processes, such as respiration. Excessive amounts of ROS can directly damage skin cells by triggering mitochondrial DNA damage or protein carbonylation; triggering skin aging can be considered alongside programmed aging. The aging process can be accelerated by environmental factors such as solar radiation, leading to premature skin aging, also known as photoaging. Solar light consists of ultra-violet (UV) radiation, visible light, and infrared radiation (IR); particularly UV radiation has detrimental effects on the skin, causing sunburn, hyperpigmentation, and inflammation through a multi-step process influenced by ROS generation [16,17]. IR radiation can have both therapeutic and pathological effects on the skin, accelerating skin aging through multiple mechanisms [18]. Visible light affects the skin in various ways, from triggering photosensitivity reactions to influencing pigmentation and wound healing [19]. Skin photoaging, a hallmark of chronic UV exposure, manifests as a complex interplay between cellular damage and disrupted tissue homeostasis. In recent years, exosomes, nanosized extracellular vesicles secreted by various cell types, including skin cells, emerged as potent modulators in photoaging [20].

UV radiation exposure acts as a potent stimulus for EVs’ biogenesis and release from epidermal cells [21,22]. This phenomenon transcends the simple act of increased exosome production. The released exosomes, brimming with bioactive molecules, actively participate in orchestrating intercellular signaling pathways within the skin microenvironment. These signaling pathways, in turn, influence various biological processes critical for skin health and function. A study showed that UVB radiation strongly stimulates the generation and release of exosomes from keratinocytes. It provided compelling evidence that platelet-activating factor receptor plays a crucial role in UVB-mediated EVs’ formation, which suggests a specific signaling pathway activated by UVB that leads to enhanced exosome secretion from keratinocytes [23]. Bioinformatics analysis dissects signaling pathways activated in keratinocytes upon exposure to EVs purified from UVA-irradiated melanocytes. The study revealed that EVs significantly enhance the proliferative and migratory capacity in keratinocytes through the upregulation of signaling mediated by the transforming growth factor beta (TGF-β1) and the inflammatory marker interleukin-6 (IL-6), ultimately affecting miR21 levels and promoting cell proliferation and migration in keratinocytes [24].

The majority of skin photoaging research traditionally used artificial light sources, which have limitations compared to natural sunlight [25,26]. Recent advancements in solar simulation technology offer a more accurate approach [27]. However, knowledge gaps persist, particularly regarding the interaction between sunlight exposure and oxidative stress in skin aging. This study aimed to detect and identify potential miRNA biomarkers expressed in exosomes derived from human epithelial adult primary keratinocytes (HEKa) in response to acute oxidative stress induced by sun solar simulator irradiation rather than UV radiation alone.

Our study employed a full-spectrum solar simulator including UV radiation, visible light, and infrared radiation to investigate the broader effects of sunlight on human keratinocytes, beyond the well-established effects of UV radiation, more specifically, their effect on microRNAs’ cargo in exosomes. By replicating the full spectrum of natural sunlight, we aimed to provide a more accurate and physiologically relevant understanding of the complex interplay between various wavelengths and their impact on human keratinocytes. We hypothesized that acute oxidative stress induced by sun solar simulator irradiation alters the expression profile of exosomal miRNAs in HEKa.

## 2. Results

### 2.1. Time-Dependent Increase in Intracellular ROS Levels of Irradiated HEKa Cells

To evaluate the temporal changes in the ROS level of HEKa cells in response to the solar simulator, the true relative ratio was measured at two time points: immediately after solar simulator irradiation and 24 h post-solar simulator irradiation for each exposure duration (10, 20, 30, 40, 50, and 60 min).
True Relative Ratio = (Fluorescence Intensity of irradiated HEKa-Background)/(Fluorescence Intensity of Control-Background)(1)

Results indicated a progressive increase in the true relative ratios immediately after solar simulator irradiation with increasing exposure time intervals, indicating a corresponding increase in the production of ROS generation within HEKa cells. Figure 1 (green bars) shows the true relative ratio of fluorescence intensity (CellROX) of HEKa cells immediately after the solar simulator irradiation at different exposure times. Furthermore, results indicated a progressive increase in the true relative ratios 24 h post-solar simulator irradiation with increasing exposure time intervals, indicating a corresponding increase in the production of ROS generation within HEKa cells, as shown in Figure 1 (violet bars).

### 2.2. Decrease in HEKa ROS Levels over Time After Solar Simulator Irradiation

A comparative analysis between the true relative ratio of fluorescence intensity of HEKa cells immediately after solar simulator irradiation and 24 h of each exposure time showed a reduction in ROS levels 24 h post-irradiation measurements compared to that immediately post-irradiation. The decrease in ROS levels 24 h after irradiation indicated that HEKa cells exhibited an activated adaptive response to mitigate the initial oxidative stress caused by solar simulator irradiation. This finding is consistent with previous studies showing the adaptive response of keratinocytes to oxidative stress induced by UVB exposure [28,29]. The most pronounced decrease in ROS levels was seen at the 40 min exposure time, indicating that this duration may be optimal for inducing a protective hormetic response, as shown in Figure 1, which represents the true relative ratio of fluorescence intensity of HEKa immediately and 24 h post-solar simulator irradiation at different exposure times.

According to paired *t*-test analysis, the average of the true relative ratio of fluorescence intensity of HEKa cells 24 h post-solar simulator irradiation was 8.1592 degrees lower than the average of the true relative ratio of fluorescence intensity of HEKa cells immediately after solar simulator irradiation; t_5_ = −3.98162 and *p*-value = 0.0053. True relative ratio = (fluorescence intensity of irradiated HEKa-Background)/(fluorescence intensity of Control-Background). Abbreviations: NAC: N-acetylcysteine, TBHP: tert-butyl hydroperoxide, ROS: reactive oxygen species. Image was created using BioRender.com. Green bars: True relative ratio of fluorescence intensity of HEKa cells immediately after the solar simulator irradiation at different exposure times; violet bar: true relative ratio of fluorescence intensity (CellROX) of HEKa cells 24 hr post the solar simulator irradiation at different exposure times. Statistical analysis of the true relative ratio of fluorescence intensity of HEKa cells immediately and 24 h post the solar simulator irradiation across all exposure times revealed a normal distribution based on the Q-Q plot, with a linear relationship and polynomial fit degree = 2, respectively. The models displayed a good fit (R-square = 0.967) and (R-square = 0.992) with a strong positive relationship between the two variables (correlation = 0.983, *p*-value = 0.0004) and (correlation = 0.925, *p*-value = 0.008). The models were statistically significant (F(1,4) = 119.72 and *p*-value = 0.0004) and (F(2,3) = 191.13 and *p*-value = 0.0007), respectively. Q-Q plot analysis is shown in Appendix A.

### 2.3. Forty-Minute Exposure to Solar Simulator Irradiation Triggers an Adaptive Response in HEKa Cells

Fold change analysis was used to identify potential HEKa adaptation to the acute oxidative stress response after solar simulator irradiation. A significant reduction in fold change, indicating the onset of an adaptive response characterized by an initial increase in ROS after irradiation, was followed by a return to baseline levels within 24 h. The most significant reduction in ROS levels (fold change of −0.747) occurred at the 40 min exposure time (as shown in Table 1).
Fold change = (true relative ratio of fluorescence intensity of HEKa cells 24 h post-irradiation − true relative ratio of fluorescence intensity of HEKa cells immediately post-irradiation)/(true relative ratio of fluorescence intensity of HEKa cells immediately post-irradiation)(2)

In conclusion, a 40 min exposure to a solar simulator was found to optimally induce acute oxidative stress. Therefore, the expression profile of exosomal miRNAs in HEKa cells was examined following a 40 min solar simulator irradiation.

An analysis of the fold change in the true relative ratio of fluorescence intensity across all exposure times revealed a normal distribution based on the Q-Q plot, with polynomial fit degree = 2. The model displayed a good fit (R-square = 0.867) and was statistically significant (F(2,3) = 9.80 and *p*-value = 0.0405).

### 2.4. Solar Simulator Irradiation Induces a Time-Dependent Decrease in Cell Survival in HEKa Cells

The percentage of cell viability was calculated; results indicated a progressive increase in the true relative ratios 24 h post-solar simulator irradiation with increasing exposure time intervals, indicating a corresponding increase in the production of ROS generation within HEKa cells (as shown in Figure 1, with polynomial fit degree = 2). The model displayed a good fit (R-square = 0.992) with a strong positive relationship between the two variables (correlation = 0.925, *p*-value = 0.008). The model was statistically significant (F(2,3) = 191.13 and *p*-value = 0.0007). Therefore, the model with a *p*-value of 0.0007 was statistically significant in predicting the true relative ratio of fluorescence intensity of HEKa cells 24 h post-solar simulator irradiation based on the equation:True Relative Ratio (24 h post-irradiation) = −7.403738 + 0.3350229 × Exposure Time + 0.0091061 × (Exposure Time − 35)^2^

In summary, the correlation between the increasing duration of exposure to the solar simulator (10, 20, 30, 40, 50, 60 min) and the Heka cells’ ROS levels was represented by the true relative ratio of fluorescence intensity (CellROX). As shown in Figure 2, it was evident that increasing the duration of exposure led to an increase in the ROS level, whether measured immediately or 24 h after the cessation of solar radiation exposure for each exposure interval.

According to the paired *t*-test analysis, the cell viability of HEKa cells 24 h post the solar simulator irradiation was 17.386 degrees lower than that of HEKa cells immediately after solar simulator irradiation; t_5_ = −6.25542 and *p*-value = 0.0008.

### 2.5. Characterization of HEKa-Derived Exosomes

#### 2.5.1. SEM Analysis of HEKa-Derived Exosome Morphology and Size Distribution

HEKa-derived exosome morphology and size distribution were analyzed using a Quanta FEG-250 scanning electron microscope following fixation with 2.5% glutaraldehyde. The micrographs, as shown in Figure 3, revealed a relatively uniform cup-shaped morphology of the HEKa-derived exosome with a size range of 40–100 nm.

#### 2.5.2. Solar Simulator Radiation-Induced Enlargement of HEKa-Derived Exosomes

A dynamic light scattering analysis revealed a significant increase in the size of exosomes isolated from HEKa cells exposed to 40 min of solar simulator irradiation compared to control cells. While control samples exhibited a single peak at 92.44 nm, irradiated samples displayed a peak at 104 nm, indicating a substantial shift toward a larger exosomal size distribution (as shown in Figure 4). These findings suggest solar radiation influences exosomal biogenesis or release processes in HEKa cells.

### 2.6. Solar Simulator Irradiation Induces Unique Exosomal miRNA Expression in HEKa

In HEKa cells, 59 exosomal miRNAs were found differentially expressed with at least a two-fold change difference with *p* < 0.05 in HEKa-irradiated cells compared to non-irradiated controls. The *p* values were calculated based on a Student’s *t*-test of three biological replicates: 2^−ΔCt^ values for each exosomal miRNA in the control group and each test group. Compared with differentially expressed exosomal miRNAs derived from the two groups (control vs. irradiated HEKa), 28 exosomal miRNAs were upregulated in irradiated HEKa, as shown in Table 2, whereas 27 miRNAs were downregulated in irradiated HEKa, as shown in Table 3. Four exosomal miRNAs were unchanged in the control group compared to the irradiated group, as shown in Table 4. Notably, novel exosomal miRNA expression patterns in irradiated HEKa cells were identified, including hsa-miR-425-5p, hsa-miR-181b-5p, hsa-miR-196b-5p, hsa-miR-376c-3p, and hsa-miR-15a-5p. Figure 5 shows the heatmap of the hierarchical clusters with the differential expression of exosomal miRNAs in non-irradiated and 40-minute irradiated HEKa cells.

### 2.7. Bioinformatic Analysis of Exosomal miRNA Targets

Target Scan Human (TargetScan) was used as a bioinformatic database to predict potential mRNA targets of differentially expressed exosomal miRNAs. Given the critical roles of proliferation, adhesion, terminal differentiation, and cornification in maintaining the cellular homeostasis of keratinocytes under oxidative stress, we focused on identifying target genes associated with these processes. The predicted target genes and their corresponding miRNA binding sites are presented in Table 5 and Table 6, respectively.

## 3. Discussion

Regarding the temporal changes in the ROS level of HEKa cells in response to solar simulator, results are consistent with existing studies, which consistently demonstrate a dose-dependent relationship between UV radiation exposure and oxidative stress in keratinocytes [30,31,32,33]. A recent study suggests a synergistic effect among solar UV, visible, and IR light components in inducing cellular damage, potentially explaining the progressive increase in ROS generation observed in our study with prolonged exposure [34].

A comparative analysis suggests that the cytotoxic effects of solar simulator irradiation on HEKa cells may not be immediately apparent. This finding is consistent with previous research demonstrating that exposure to simulated solar radiation can induce immediate alterations in the DNA structure of the skin, while changes in protein signatures manifest at a later stage with significant changes in protein phosphorylation observed up to 24 h post-irradiation. This temporal discrepancy highlights the delayed cytotoxic response of skin cells to solar radiation, suggesting a gradual process of cellular damage and dysfunction [27,35].

Radiation-induced exosome size and composition alterations represent a complex interplay of cellular stress responses. Potential mechanisms include disruptions to the intricate machinery governing exosome biogenesis, leading to variations in size distribution. Additionally, the cargo selectively packaged within exosomes might be modified following irradiation, influencing overall particle properties. Lastly, changes in the lipid composition of the cell membrane, a consequence of radiation-induced stress, could indirectly affect exosome size and structure. While these pathways are plausible, definitive evidence establishing a direct causal relationship between radiation and specific alterations in exosome biogenesis and release remains elusive and warrants further investigation. Exosomes as key mediators of intercellular communication undergo marked alterations in response to the effects of different types of radiation, including solar radiation on exosome size, and are not explicitly discussed in the provided contexts; direct evidence linking irradiation to specific changes in cells-derived exosome size remains limited. Researchers emphasize the impact of radiation on exosome function and content rather than their physical dimensions. Studies on diverse cell types including prostate cancer, pancreatic carcinoma, and mesenchymal stem and melanocyte cells consistently demonstrated that irradiation induces variable alterations in exosomal cargo and content. A strong correlation between these radiation-induced changes in exosomal composition and cellular stress responses suggests a potential link to modifications in exosomal dimensions [36,37].

Significant differential expression in exosomal miRNAs derived from keratinocytes in response to solar simulator irradiation supports our research hypothesis that acute oxidative stress from solar simulator irradiation alters the expression of exosomal miRNAs in human adult epidermal keratinocytes. Notably, several of the differentially expressed miRNAs identified in this study, including hsa-miR-22, hsa-let-7b, hsa-miR-125b, hsa-miR-24, hsa-miR-27b, hsa-miR-21-3p, hsa-miR-200, and hsa-miR-203, were previously implicated in regulating keratinocyte proliferation, differentiation, and senescence [6,38,39]. This aligns with previous studies, strengthens the results, and underscores the potential of these exosomal miRNAs in mediating the cellular response of HEKa cells to oxidative stress.

Exosomal microRNAs can affect cell proliferation, cell differentiation, and aging in the epidermis. The let-7 family including hsa-miR-22, hsa-miR-27b, and hsa-miR-21 is among the most abundant miRNAs in keratinocyte-derived exosomes. In addition, hsa-miR-200, hsa-miRNA-203, and hsa-miRNA-205 are detected in high expressions [13].

The hsa-miR22 plays a crucial role in regulating keratinocyte differentiation [40,41]. MiRNAs hsa-miR-125b and hsa-miR-24 are often found in keratinocyte-derived exosomes. They are involved in maintaining the multipotent properties of keratinocytes, regulating the self-renewal of stem cells and inhibiting cell differentiation by transcriptionally suppressing the B lymphocyte-induced maturation protein-1 and vitamin D receptor genes. Furthermore, the upregulated expression of hsa-miR-24 in human epidermal stem cells is known to inhibit the expression of P27 and P16 proteins by post-transcriptionally suppressing cyclin-dependent kinase to promote the G1/S transition of stem cells and inhibit stem cell differentiation [42].

On the other hand, hsa-miR-27 b was shown to suppress the bone morphogenetic protein signaling pathway, particularly phosphorylated SMAD1/5, during the final endoderm differentiation. This leads to the downregulation of me-endodermal marker genes and inhibits early differentiation processes [38]. Interestingly, hsa-miR21-3p promotes proliferation and inhibits apoptosis of keratinocytes by regulating the JAK/STAT signaling pathway and cytokeratin-17 expression [43].

hsa-miR-200 and hsa-miR-203 are preferentially expressed in the epidermis. In particular, hsa-miR-203 is 100-fold more abundant in the skin than in other tissues and is referred to as a skin-specific miRNA expressed in human epidermal stem cells [39]. The hsa-miR203 promotes epidermal cell differentiation by controlling cell proliferation potential and inducing cell cycle arrest, suggesting that hsa-miR-203 serves as a switch for proliferation and differentiation in skin development. Furthermore, has-miR-203 inhibits the long-term proliferation of progenitor cells by targeting key regulators of cell cycle and cell division, such as p63. It regulates interleukin-8 (IL-8) via two predicted hsa-miR-203 binding sites in the 3′ UTR of IL-8, thereby suppressing IL-8 mRNA and protein expression in primary human keratinocytes [6,39].

Comprehensive bioinformatics analysis identified the target genes of the upregulated miRNAs, including hsa-miR-125b-5p, hsa-miR-184, hsa-miR-24-3p, hsa-miR-205-5p, and hsa-miR-376a-3p, which are all associated with human epidermal stem cells’ proliferation. In contrast, target genes of the downregulated hsa-miR-210 and hsa-miR-483-3p are associated with the proliferation of human epidermal stem cells. The differentiation of human epidermal stem cells is associated with the downregulation of hsa-let-7b-3p, hsa-miR-106a, hsa-miR-203, hsa-miR-210, hsa-miR-23b-3p, hsa-miR-31, hsa-miR-328, hsa-miR-34a-3p, hsa-miR-574-3p, and hsa-miR-720 [44].

The present study employed the Human miFinder Focus PCR Panel for miRNA profiling; while it provided valuable data, it was restricted to only 84 miRNAs. A more comprehensive assessment of exosomal miRNA profiles would necessitate the use of broader profiling techniques such as microarrays, enabling a wider interrogation of exosomal microRNA species. Furthermore, we evaluated the impact of solar-simulated radiation on epidermal keratinocytes using ROS production as a biomarker of cellular damage; a more comprehensive assessment would involve the inclusion of additional biomarkers like mitochondrial DNA and nuclear DNA damage. Incorporating these additional markers would provide a more robust understanding of the cellular responses to solar radiation.

To further clarify the validation of the predicted target genes and their corresponding exosomal miRNA binding sites, we recommend employing the dual-luciferase reporter assay. It is a powerful technique to validate miRNA target interactions within the 3′ UTR of a gene. This technique involves cloning a target gene’s 3′ UTR downstream of a luciferase reporter, co-transfecting it with the miRNA of interest into HEKa cells and measuring luciferase activity. A control construct with a mutated 3′ UTR lacking the miRNA binding site is used to normalize luciferase activity. By comparing luciferase activity between wild-type and mutant constructs, we can determine whether a specific miRNA directly targets the 3′ UTR of a gene and represses its expression. If the miRNA targets the 3′ UTR, it will bind and inhibit luciferase expression.

In future work, we could employ synthetic pre-miR miRNA precursor (pre-miRs) molecules or anti-miR miRNA (anti-miRs) inhibitors to elucidate further the functional roles of differentially expressed exosomal miRNAs identified in HEKa cells exposed to solar simulator irradiation. They can manipulate miRNA levels and assess their impact on cellular responses to oxidative stress. Pre-miRs are double-stranded RNA molecules that mimic their endogenous mature miRNA and increase the level of mature miRNAs in the cell. As the level of mature miRNAs rises, a decrease in the expression of their target mRNA is expected. Precursor negative control has a random sequence pre-miR molecule, similar to pre-miRs but the pre-miR-control produces a random scrambled miRNA with no effect on targets. This makes the pre-miR-control an ideal negative control compared to any other small RNA fragment, whereas it is possible to commercially obtain pre-miR for transfection into HEKa cells. Anti-miRs are synthetic oligonucleotides designed to inhibit the function of mature miRNAs specifically. These molecules function by binding to the miRNA sequence, preventing its incorporation into the RNA-induced silencing complex and thereby blocking its ability to target and regulate gene expression.

Commercially available pre-miRs and anti-miRs, such as LNA miRNA inhibitors and HDO-anti-miR technologies, can further facilitate the application of these approaches in future studies. Through the integration of these tools with advanced analytical techniques, it is expected that a more profound comprehension of the regulatory pathways governing exosomal miRNA-mediated responses to oxidative stress in HEKa cells will be attained, facilitating the development of novel therapeutic strategies.

## 4. Materials and Methods

### 4.1. Human Keratinocytes’ Cell Line

Human primary epidermal keratinocytes and normal, adult cells (HEKa) were purchased from the American Type Culture Collection (ATCC, cat. no. PCS-200-011™) (Manassas, VA, USA), in addition to dermal basal medium with extract P: 0.4%; rh TGF-alpha: 0.5 ng/mL; L-glutamine: 6 mM; hydrocortisone: 100 ng/mL; insulin: 5 µg/mL; epinephrine: 1.0 µM; Apo-transferrin: 5 µg/mL; and 1% gentamicin/amphotericin solution to create a complete serum-free culture environment for keratinocytes (ATCC, cat. no. PCS-200-030).

### 4.2. Primary HEKa Cell Culture

Primary HEKa cells were cultured in a humidified 37 °C, 5% CO_2_, incubator. The optimal seeding density for HEKa cell proliferation was determined to be 2500 cells/cm^2^. The culture medium was changed to a freshly supplemented medium, 24 h after establishing a culture from cryopreserved cells. The medium was changed every other day thereafter until the culture was approximately 50% confluent. Once the culture reached 50% confluence, medium was changed every day until the culture was approximately 70% confluent. To passage HEK cells, we used a T75 flask with 70% confluency and 6 × 10^6^ cells. We added TrypLE Express Enzyme solution (Gibco, cat. no.12604013) (Waltham, MA, USA) and then deactivated it with PBS. We centrifuged the cell suspension, discarded the supernatant, and resuspended the cell pellet in a complete growth medium. For cryopreservation, we resuspended the cell pellet in Synth-a-Freeze Cryopreservation medium (Gibco, cat. no. A1254201), aliquoted into cryovials, and froze at −80 °C before transferring to a liquid nitrogen tank for long-term storage. The concentration of viable cells in 1 ml cell suspension was determined using the trypan blue dye exclusion. Ten µL of cell suspension was mixed with 10 µL of trypan blue stain (0.4%) for use with an automated cell counter (Invitrogen, cat. no. T10282) (Waltham, MA, USA). Ten µL of the stained cell suspension was transferred to the (V) groove on both sides of the LUNA™ cell counting slides to be counted with the LUNA™-II automated cell counter. All culture flasks were seeded at optimal density for all experiments (2500 cells/cm^2^).

### 4.3. Solar Irradiation of HEKa ROS Level Measurements

The solar irradiator used in this study was the ABET Technology Sun2000 (High Output Model). It is specifically designed to replicate global solar irradiance. The distance between the specimens and source was standard [45]. To investigate the impact of acute oxidative stress on ROS levels, HEKa cells were exposed to varying durations (10, 20, 30, 40, 50, and 60 min) of irradiation. Three independent cell cultures in T75 flasks were used as biological replicates for each time point exposure in this experiment. The solar simulator was pre-warmed for 30 min as recommended by the manufacturer. HEKa cells were irradiated in phosphate-buffered saline (PBS) for all experiments. This approach prevented the absorption of UV radiation by the culture medium, which could reduce the effective dose reaching the cells and eliminate the potential for phototoxicity arising from oxidized organic components present in a culture medium [46]. The trypsinized cells were resuspended in their respective complete growth media (~45 mL per flask). Subsequently, 1 mL from each cell suspension was seeded into separate 35 mm Petri dishes with ~5 × 10^5^ cells/mL, resulting in a total of six dishes per exposure time point

### 4.4. ROS Level Measurements

Detection of total intracellular ROS was performed using the CellROX Deep Red Flow Cytometry Assay Kit (Invitrogen, cat. no. C10491) according to the manufacturer’s instructions. Three independent biological replicates were performed, each involving technical replicates to enhance measurement reliability. ROS level measurements were assessed at two time points: immediately after irradiation and 24 h post-irradiation for each exposure duration. A return of ROS levels to baseline within a 24 h timeframe was indicative of an acute oxidative stress response [28].

A cell suspension of 5 × 10^5^ cells/mL for each sample was adjusted in a complete medium in a 1.5 mL Eppendorf tube. The desired final concentration of CellROX Deep Red stain was [500 nM]. The reagent was added to both controls and samples and incubated under normal growth conditions at 37 °C, 5% CO_2_, protected from light, and wrapped with aluminum foil for 30 min. For CellROX Deep Red assay controls, negative control cells were incubated with [250 mM] NAC per 1 mL of cell suspension for a final concentration of [5 mM] for 1 h under normal growth conditions at 37 °C, 5% CO_2_. Following 1 h incubation with NAC, positive control cells were incubated with [50 mM] TBHP per 1 mL of cell suspension for a final concentration of [200 µM] for 30 min under normal growth conditions at 37 °C, 5% CO_2_. After that, 100 µL of controls and samples was transferred to a black-walled, clear-bottom, 96-well container. The fluorescence was detected at 635 nm excitation and 665 nm emission on a BioTek Synergy H1 microplate reader (Winooski, VT, USA) and analyzed using Biotek gene 5 data analysis software, version 3.05.11. Three independent biological replicates, each with three technical replicates, were performed. The fluorescence intensity was measured at an excitation wavelength of 635 nm and an emission wavelength of 665 nm, and we subsequently calculated the true relative ratio for each sample. The true relative ratio represents the fluorescence intensity of irradiated HEKa cells relative to the background and control samples to provide a normalized measure of ROS levels.

### 4.5. Cell Viability: MTT Assay

Cell viability was assessed using the colorimetric MTT (3-(4,5-dimethylthiazol-2-yl)-2,5-diphenyltetrazolium bromide) assay following exposure to ABET Technology Sun2000 solar simulator for varying durations (10, 20, 30, 40, 50, and 60 min). Three independent biological replicates were performed, with technical replicates within each biological replicate to enhance data reliability. Cells were seeded at 10^4^ cells/well in 96-well flat-bottomed plates and allowed to recover for 24 h at 37 °C in a humidified 5% CO_2_ incubator. The culture medium was carefully removed. Cells were then irradiated in 100 μL of DPBS, with the lid open. Control wells were covered with aluminum foil. After irradiation, PBS was removed and replaced with 200 μL of complete medium. Cell viability was assessed at two time points: immediately and 24 h post-irradiation for each exposure duration. Briefly, 20 μL of MTT solution (10 mg/mL DPBS) was added to each well and the plates were incubated for 4 h at 37 °C followed by the removal of the medium and addition of 100 μL of lysis solution-dimethyl sulfoxide DMSO to each well. Plates were covered with foil and left on a plate shaker for approximately 10 min. The plates were then read on a Biotech Synergy H1 microplate reader at 550 nm. The percentage of cell viability was calculated relative to the background and control samples that received no irradiation to provide a normalized measure at two time points: immediately after solar simulator irradiation and 24 h post-solar simulator irradiation for each exposure duration (10, 20, 30, 40, 50, and 60 min).
% Cell Viability = (Absorbance_treatment_ − Absorbance_background_)/(Absorbance_control_ − Absorbance_background_) × 100%

### 4.6. Exosome Isolation

HEKa cells were subcultured and grown in 100 mm Nunc tissue culture dishes (Thermo Fisher, Waltham, MA, USA). Control cells and irradiated cells were seeded at 2500 cells/cm^2^. Three independent biological replicates were performed, with technical replicates within each biological replicate including irradiated samples and controls. Once 60% confluence was reached, the cell culture medium was replaced with fresh medium and incubated at 37 °C in a 5% CO_2_ incubator. HEKa-derived exosome isolation was performed using an exoEasy Maxi kit (Qiagen, cat. no. 76064), according to the manufacturer’s instructions. After 48 h of incubation, the conditioning medium from samples and controls was collected, filtered, and loaded into the exoEasy spin column. The samples were then stored at 4 °C for further exosome characterization.

### 4.7. Exosome Characterization

The size distribution and morphology of HEKa-derived exosomes were characterized using dynamic light scattering DLS with a Zetasizer 90 (Malvern Panalytical, Malvern, UK) and scanning electron microscopy analysis with Quanta FEG-250 SEM (FEI Company, Hillsboro, OR, USA), respectively, at the Nanotechnology Institute, Jordan University of Science and Technology. The size of HEKa-derived exosomes was measured using a Zetasizer 90 for better detection; the exosome samples and controls were diluted with sterile-filtered PBS at a dilution factor of 1:1000. Then, 1 mL of diluted exosome samples and controls was loaded into a polystyrene, square, disposable cuvette that was pre-rinsed with filtered ultrapure water before introducing the sample. The temperature and sitting measurements of the cuvette holder were equilibrated as 25 °C, 1.45 for the reflective index, and 0.001 for the absorption index, before starting analysis using the Zetasizer 90 software, version 7.11. HEKa-derived exosomes were imaged using a Quanta FEG-250 SEM. First, 0.5 mL of diluted exosome samples was fixed with an equal volume of pre-prepared 2.5% glutaraldehyde solution obtained by diluting 1 mL of 25% glutaraldehyde with 10 mL of distilled water. The fixed samples were then mounted onto SEM stubs. Subsequently, the samples mounted on the SEM stub were left to dry overnight at −4 °C. Finally, to enhance conductivity and prevent charging during SEM imaging, the exosomes were sputter-coated with a thin layer of gold using a Q150R ES Quorum Technology sputter-coating system (Quorum Technology, Laughton, East Sussex, UK).

### 4.8. HEKa-Derived Exosomal miRNAs’ Extraction

HEKa cells were grown in 100 mm dishes, and three biological replicates were performed, including irradiated and control samples. Conditioned medium was collected after 48 h, and exosomes were isolated. MiRNA extraction was performed using the miRNeasy Kit (Qiagen, cat. no. 217084), and the purified miRNA eluate was stored at −80 °C for further analysis. Each measurement was performed in triplicates.

### 4.9. Reverse Transcription Reaction

For the transcription reaction, the miRCURY LNA miRNA kit (Qiagen, cat. no. 339306) was used to quantify miRNA samples and controls. It utilizes polyadenylation by poly(A) polymerase on the universal reverse transcriptase (RT) primers at the 3′ end. In qPCR, miRNA-specific primers are amplified by locked nucleic acid (LNA). The RT reaction was performed in a thermal cycler for 60 min at 41 °C, followed by 5 min at 95 °C to inactivate the RT enzyme and immediately cooled to 4 °C to stop the reaction. The resulting complementary DNA (cDNA) served as a template for subsequent quantitative real-time polymerase chain reaction (qPCR) analysis of exosomal miRNA expression.

### 4.10. Quantitative PCR (qPCR)

For miRNA expression analysis on a 96-well plate, miRCURY LNA miRNA SYBR Green qPCR Assay (Qiagen, cat. no. 339306) was used. The assay included 84 different miRNAs of Human miFinder Focus, chosen based on exosomal miRNA data from the Exocarta database. The qPCR reaction was prepared using a master mix containing 2× miRCURY SYBR Green master mix (Qiagen, Hilden, Germany), undiluted cDNA template, and RNase-free water. The reaction mixture was dispensed into the PCR panel plate and sealed. Real-time PCR was performed using the Quant Studio 5 with thermal cycling at 95 °C for 2 min, followed by 40 cycles of denaturation at 95 °C for 10 s and annealed/extended at 56 °C for 60 s. HEKa-derived exosomal miRNA expression studies were conducted at the Enzyme Company laboratories

### 4.11. The qPCR Data Analysis

Statistical analyses were performed using the GeneGlobe Data Analysis Center (Qiagen, Hilden, Germany). CT values exceeding 35 were designated as undetermined. Two criteria were established for data acceptance and analysis: (a) The entire data set was excluded if the miRNA negative control primer showed successful annealing, indicating possible sample contamination. (b) At least two of the 84 assessed microRNAs needed to exhibit stable expression between treatment groups to serve as housekeeping genes. The traditional housekeeping genes SNORD44, SNORD38B, SNORD49A, and U6 snRNA are commonly used as reference genes for normalizing miRNA levels. However, high cycle threshold (CT) values across experimental and control groups indicated inadequate reliability and expression levels of snRNAs within the exosomal miRNAs of HEKa cells. The geNorm algorithm was used to analyze all detected miRNAs, and the fold change/regulation was calculated using the delta-delta CT method.

### 4.12. Data Presentation and Statistical Analysis

The statistical analysis of ROS level measurements and MTT assay was performed using JMP16 software, version 16.2.0, whereas data visualization related to them was accomplished by generating graphs using Excel software. The data are presented as mean ± standard error of the Mean (SEM). Paired *t*-tests were used to determine statistical significance, with significance accepted at a *p*-value of <0.05. Data visualization and interpretation were accomplished by generating graphs using Excel software and the QIAGEN GeneGlobe platform.

## 5. Conclusions

The present study investigates the effects of solar simulator irradiation on the expression profile of exosomal miRNAs in HEKa cells, focusing on oxidative stress, and cell viability. Our findings demonstrate that exposure to simulated solar radiation leads to a time-dependent increase in intracellular levels of ROS in HEKa cells, in addition to a decrease in cell viability with a delayed cytotoxic response of skin cells to solar simulator radiation, suggesting a gradual process of cellular damage and dysfunction. Additionally, we observed a significant increase in the size of exosomes isolated from irradiated HEKa cells, suggesting an influence of solar simulator radiation on exosomal biogenesis or release processes. Furthermore, 59 exosomal miRNAs are identified as being differentially expressed in irradiated HEKa cells, with at least a two-fold difference in expression, in addition to novel differentially expressed exosomal miRNAs in irradiated keratinocytes, including hsa-miR-425-5p, hsa-miR-181b-5p, hsa-miR-196b-5p, hsa-miR-376c-3p, and hsa-miR-15a-5p.

## Figures and Tables

**Figure 1 ijms-25-12477-f001:**
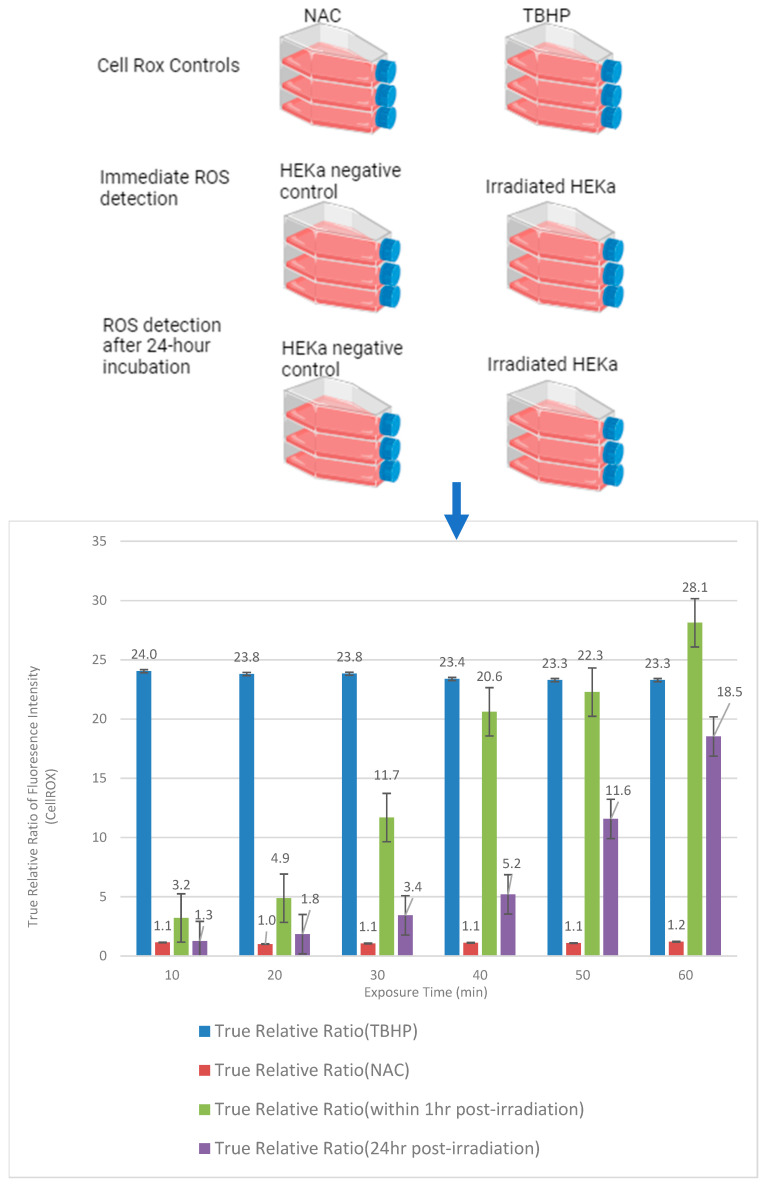
The true relative ratio of fluorescence intensity (CellROX) of HEKa cells immediately (at selected time intervals (10, 20, 30, 40, 50, 60 min)) and 24 h post-solar simulator irradiation at different exposure times. As shown in the graph, the three biological repeats for both control cells and HEKa irradiated cells were compared immediately after irradiation and at 24 h post-irradiation after the addition of NAC and TBHP as further controls. The average of the true relative ratio of fluorescence intensity of HEKa cells immediately after the solar simulator irradiation was 15.137 ± 2.049. On the other hand, the average of the true relative ratio of fluorescence intensity of HEKa cells 24 h post the solar simulator irradiation was 6.978 ± 1.663. Furthermore, both variables can be assumed normally distributed using Q-Q plots with strong correlation (correlation value = 0.896 and *p*-value = 0.0156).

**Figure 2 ijms-25-12477-f002:**
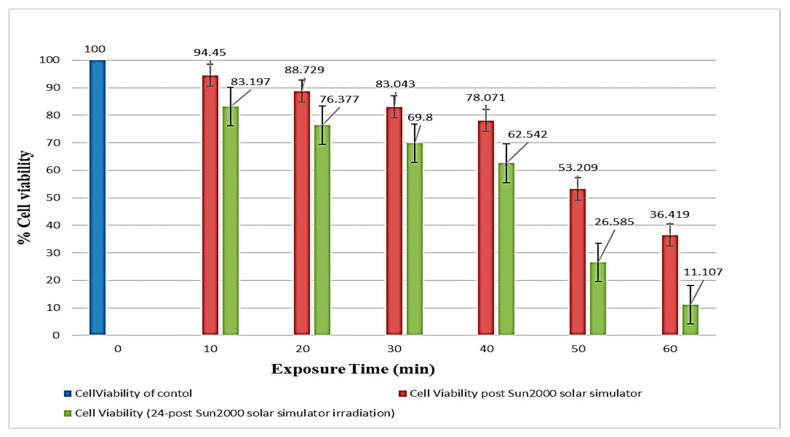
Cell viability of HEKa cells immediately and 24 h (24 h) post-solar simulator irradiation at different exposure times. A comparative analysis between the cell viability of HEKa cells immediately after solar simulator irradiation and 24 h after for each exposure time revealed a reduction in cell viability 24 h post-irradiation measurements compared to the immediate post-irradiation at different exposure times. The average cell viability of HEKa cells immediately after solar simulator irradiation was 72.320 ± 9.239, whereas the average cell viability of HEKa cells 24 h post-solar simulator irradiation was 54.934 ± 11.919. Furthermore, both variables could be assumed as normally distributed using Q-Q plots with strong correlation (correlation value = 0.997 and *p*-value < 0.0001).

**Figure 3 ijms-25-12477-f003:**
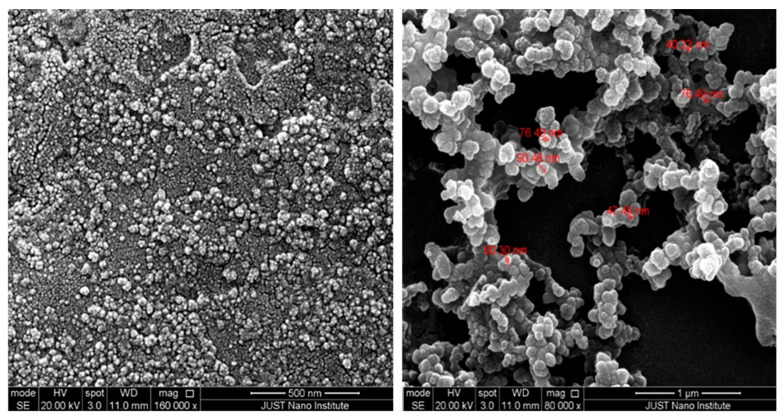
Quanta FEG-250 scanning electron micrographs of fixed HEKa-derived exosomes.

**Figure 4 ijms-25-12477-f004:**
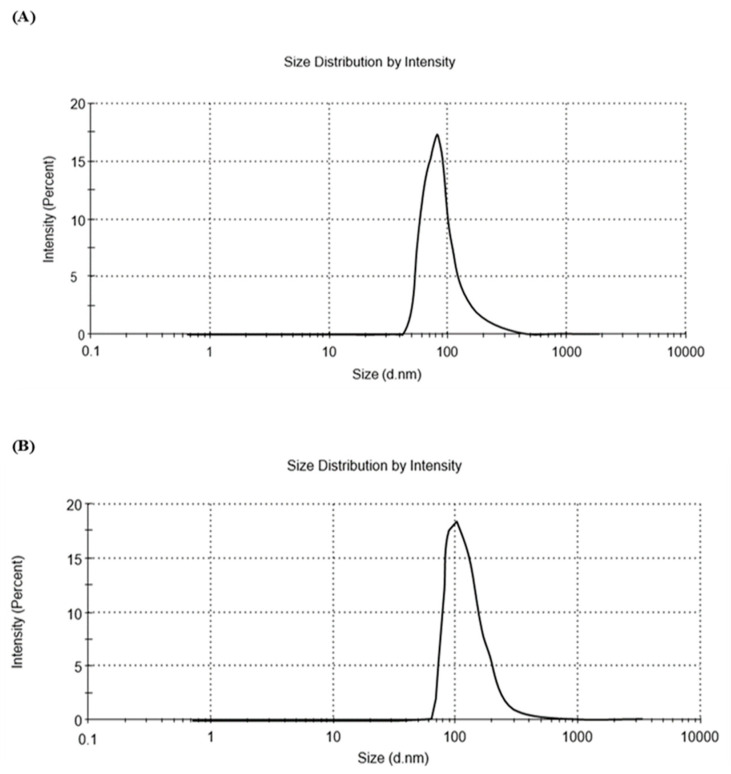
Zetasizer 90 data analysis; particle size distribution by intensity analysis of HEKa-derived exosomes. (**A**): Dynamic light scattering analysis of control samples. (**B**): Dynamic light scattering analysis of irradiated samples.

**Figure 5 ijms-25-12477-f005:**
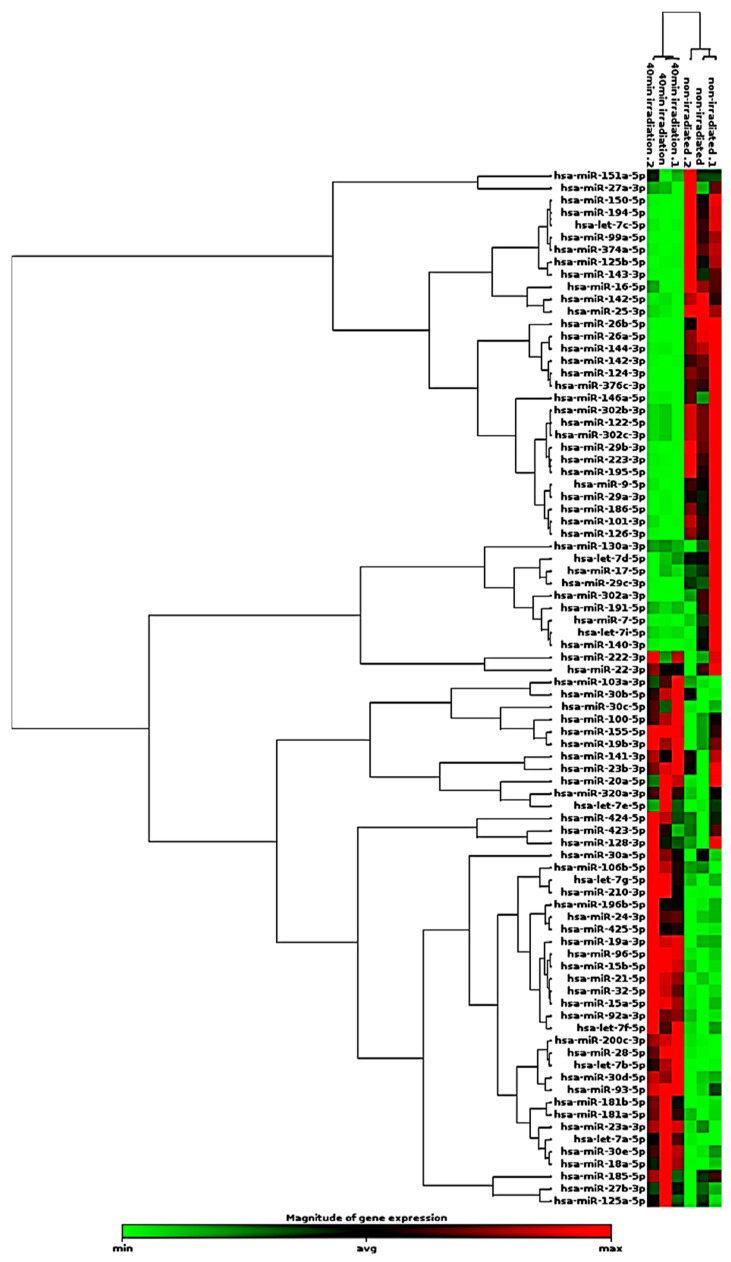
Heatmap with hierarchical clustering. The hierarchical clustering heatmap visualizes the differential expression of exosomal miRNAs in non-irradiated and 40 min irradiated HEKa cells. Each row represents a specific miRNA, and each column represents a biological sample. The color intensity of each cell corresponds to the relative expression level of a particular miRNA in a given sample. Red cells suggest upregulation in the irradiated group, while green cells indicate downregulation. The dendrograms provide insights into the hierarchical relationships among miRNAs and samples, potentially revealing miRNA families or groups with similar functions.

**Table 1 ijms-25-12477-t001:** Fold change in the true relative ratio of fluorescence intensity.

Exposure Time (min)	True Relative Ratio (Immediately)	True Relative Ratio (24 h Post-Irradiation)	Fold Change
10	3.217	1.27	−0.605
20	4.882	1.846	−0.621
30	11.694	3.438	−0.706
40	20.62	5.21	−0.747
50	22.276	11.573	−0.48
60	28.134	18.531	−0.341

**Table 2 ijms-25-12477-t002:** Exosomal miRNAs upregulated in test group vs. control group.

Position	miRNA ID	Fold Change	*p*-Value
A07	hsa-miR-103a-3p	2.49	0.044384
A10	hsa-miR-32-5p	7.78	0.001252
A12	hsa-let-7g-5p	3.46	0.006857
B01	hsa-miR-30c-5p	2.24	0.042151
B02	hsa-miR-96-5p	5.33	0.00002
B05	hsa-miR-24-3p	2.92	0.005705
B06	hsa-miR-155-5p	2.71	0.0161
B08	hsa-miR-425-5p	6.23	0.011311
B09	hsa-miR-181b-5p	4.38	0.012071
B12	hsa-miR-21-5p	5.63	0.000433
C01	hsa-miR-30e-5p	3.31	0.004399
C02	hsa-miR-200c-3p	7.37	0.000209
C03	hsa-miR-15b-5p	5.35	0.000074
C06	hsa-miR-210-3p	14.04	0.008076
C07	hsa-miR-15a-5p	3.51	0.00038
C08	hsa-miR-181a-5p	3.24	0.00543
C11	hsa-miR-28-5p	5.06	0.001059
D05	hsa-miR-19a-3p	3.88	0.000217
D06	hsa-miR-18a-5p	3.62	0.015969
D09	hsa-let-7a-5p	14.1	0.007496
D11	hsa-miR-92a-3p	3.22	0.00192
D12	hsa-miR-23a-3p	3.25	0.000909
F05	hsa-let-7b-5p	5.06	0.002922
F06	hsa-miR-19b-3p	2.7	0.046281
F08	hsa-miR-93-5p	2.9	0.001819
F10	hsa-miR-196b-5p	3.63	0.031736
G06	hsa-let-7f-5p	5.15	0.003592
G11	hsa-miR-100-5p	3.7	0.039043

**Table 3 ijms-25-12477-t003:** Exosomal miRNAs downregulated in test group vs. control group.

Position	miRNA ID	Fold Change	*p*-Value
A01	hsa-miR-142-5p	0.3	0.003278
A02	hsa-miR-9-5p	0.06	0.012246
A03	hsa-miR-150-5p	0.02	0.004472
A05	hsa-miR-101-3p	0.08	0.003293
A06	hsa-let-7d-5p	0.41	0.043286
A09	hsa-miR-26a-5p	0.02	0.000362
A11	hsa-miR-26b-5p	0.03	0.004581
B04	hsa-miR-142-3p	0.04	0.003394
B10	hsa-miR-302b-3p	0.43	0.001049
C04	hsa-miR-223-3p	0.08	0.002307
C05	hsa-miR-194-5p	0.03	0.003567
C09	hsa-miR-125b-5p	0.11	0.009593
C10	hsa-miR-99a-5p	0.03	0.003008
D02	hsa-miR-29b-3p	0.03	0.001329
D03	hsa-miR-29a-3p	0.07	0.01861
D07	hsa-miR-374a-5p	0.05	0.001323
D10	hsa-miR-124-3p	0.02	0.002044
E01	hsa-miR-25-3p	0.16	0.000285
E03	hsa-miR-376c-3p	0.01	0.004087
E04	hsa-miR-126-3p	0.04	0.003744
E05	hsa-miR-144-3p	0.18	0.000505
E10	hsa-miR-195-5p	0.01	0.003866
E11	hsa-miR-143-3p	0.05	0.017572
F09	hsa-miR-186-5p	0.02	0.010657
G02	hsa-let-7c-5p	0.05	0.002435
G07	hsa-miR-122-5p	0.43	0.001049
G12	hsa-miR-302c-3p	0.43	0.001049

**Table 4 ijms-25-12477-t004:** Exosomal miRNAs unchanged in test group vs. control group.

Position	miRNA ID	Fold Change	*p*-Value
A08	hsa-miR-16-5p	0.59	0.003229
B11	hsa-miR-30b-5p	1.51	0.03102
E12	hsa-miR-30d-5p	1.85	0.000927
G09	hsa-miR-106b-5p	1.9	0.010788

**Table 5 ijms-25-12477-t005:** Targets of exosomal miRNAs expressed in HEKa cells, which were either down- or upregulated in response to 40 min of solar simulator irradiation, using TargetScan as mRNA/miRNtarget database.

Gene Symbol	Gene Product	Physiological Process	Upregulated Exosomal miRNAs	Downregulated Exosomal miRNAs
*CDKN1A*	Cyclin-dependent kinase inhibitor 1	keratinocyte proliferation	hsa-miR-93-5p, hsa-let-7b-5p, hsa-let-7g-5p, hsa-let-7a-5p, hsa-let-7f-5p	hsa-let-7d-5p, hsa-let-7c-5p
*IRF6*	Interferon regulatory factor 6	keratinocyte proliferation	hsa-miR-96-5p	
*FERMT1*	FERM domain containing kindlin 1	keratinocyte proliferation	hsa-miR-24-3p	
*FST*	Follistatin	keratinocyte proliferation	hsa-miR-24-3p, hsa-miR-96-5p, hsa-miR-425-5p, hsa-miR-30c-5p, hsa-miR-30e-5p, hsa-miR-92a-3p, hsa-miR-32-5p	hsa-miR-144-3p, hsa-miR-25-3p
*WNT16*	Protein Wnt-16	keratinocyte proliferation	hsa-miR-103a-3p, hsa-miR-425-5p	
*OVOL1*	Putative transcription factor Ovo-like 1	keratinocyte proliferation	hsa-miR-96-5p, hsa-miR-30c-5p, hsa-miR-30e-5p	hsa-miR-125b-5p
*NFKBIZ*	NF-kappa-B inhibitor zeta	keratinocyte proliferation		hsa-miR-374a-5p
*FGF10*	Fibroblast growth factor 10	keratinocyte proliferation	hsa-miR-425-5p, hsa-miR-19a-3p, hsa-miR-19b-3p	hsa-miR-302b-3p, hsa-miR-9-5p
*EREG*	Proepiregulin	keratinocyte proliferation	hsa-miR-19b-3p, hsa-miR-19a-3p, hsa-miR-93-5p	
*PTCH1*	Protein patched homolog 1	keratinocyte proliferation	hsa-miR-15b-5p, hsa-miR-15a-5p, hsa-miR-200c-3p	hsa-miR-9-5p
*CDH13*	Cadherin-13	keratinocyte proliferation	hsa-miR-30e-5p, hsa-miR-30c-5p	
*SDR16C5*	Epidermal retinol dehydrogenase 2	keratinocyte proliferation		hsa-miR-194-5p
*IVL*	Involucrin	keratinocyte terminal differentiation	hsa-miR-200c-3p	hsa-miR-26a-5p
*KRT10*	Keratin 10	keratinocyte terminal differentiation	hsa-miR-93-5p	hsa-miR-186-5p
*CDH1*	Cadherin 1	keratinocyte adhesion		hsa-miR-9-5p
*DSG1*	Desmoglein 1 (Desmocollin)	keratinocyte adhesion	hsa-miR-23a-3p, hsa-miR-210-3p, hsa-miR-32-5p, hsa-miR-92a-3p, hsa-miR-24-3p, hsa-miR-9-5p, hsa-miR-26a-5p	
*ITGA3*	Integrin subunit alpha 3	keratinocyte adhesion	hsa-miR-181a-5p, hsa-miR-181b-5p	
*CERS3*	Ceramide synthase 3	keratinocytes’ cornification		hsa-miR-150-5p
*CYP26B1*	Cytochrome P450 family 26 subfamily B member 1	keratinocytes’ cornification	hsa-miR-100-5p	hsa-miR-302c-3p
*KLK5*	Kallikrein-5	keratinocytes’ cornification	hsa-miR-103a-3p,	hsa-miR-122-5p, hsa-miR-143-3p
*FLG*	Filaggrin	keratinocyte differentiation marker		hsa-miR-125b-5p

**Table 6 ijms-25-12477-t006:** Predicted consequential pairing between target regions and exosomal miRNAs of HEKa cells.

3′ UTR Target	Predicted Consequential Pairing of Target Region (Top) and miRNA (Bottom)	Site Type	Context++ Score
Position 468–474 of CDKN1A 3′ UTRhsa-miR-93-5p	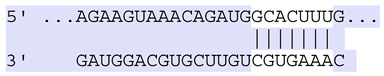	7mer-m8	−0.20
Position 943–950 of CDKN1A 3′ UTRhsa-let-7b-5p	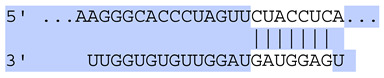	8mer	−0.44
Position 943–950 of CDKN1A 3′ UTRhsa-let-7g-5p	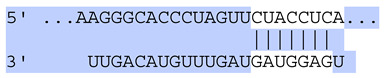	8mer	−0.43
Position 943–950 of CDKN1A 3′ UThsa-let-7a-5p	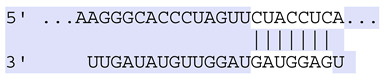	8mer	−0.43
Position 943–950 of CDKN1A 3′ UTRhsa-let-7d-5p	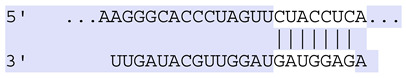	8mer	−0.46
Position 943–950 of CDKN1A 3′ UTRhsa-let-7c-5p	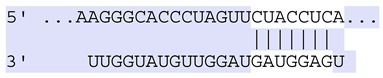	8mer	−0.43
Position 640–647 of IRF6 3′ UTRhsa-miR-96-5p	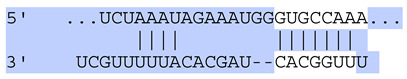	8mer	−0.22
Position 1573–1580 of FERMT1 3′ UTRhsa-miR-24-3p	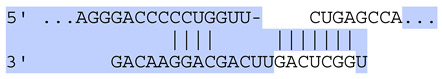	8mer	−0.12
Position 16–23 of FST 3′ UTRhsa-miR-24-3p	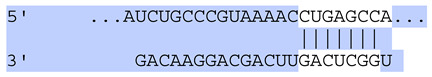	8mer	−0.65
Position 108–114 of FST 3′ UTRhsa-miR-96-5p	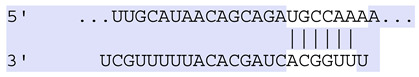	7mer-A1	−0.21
Position 611–617 of FST 3′ UTRhsa-miR-425-5p	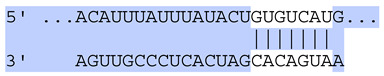	7mer-m8	−0.29
Position 916–923 of FST 3′ UTRhsa-miR-30c-5p	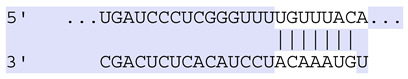	8mer	−0.22
Position 916–923 of FST 3′ UTRhsa-miR-30e-5p	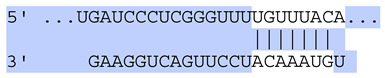	8mer	−0.22
Position 1368–1374 of FST 3′ UTRhsa-miR-92a-3p	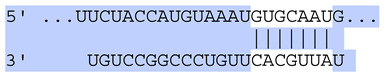	7mer-m8	−0.31
Position 1368–1374 of FST 3′ UTRhsa-miR-32-5p	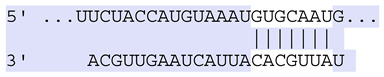	7mer-m8	−0.33
Position 606–612 of FST 3′ UTRhsa-miR-144-3p	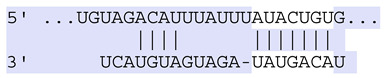	7mer-m8	−0.36
Position 1368–1374 of FST 3′ UTRhsa-miR-25-3p	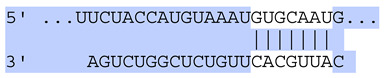	7mer-m8	−0.31
Position 1372–1379 of WNT16 3′ UTRhsa-miR-103a-3p	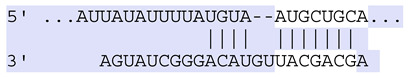	8mer	−0.46
Position 665–671 of OVOL1 3′ UTRhsa-miR-96-5p	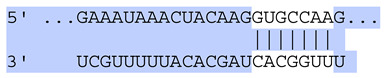	7mer-m8	−0.30
Position 1761–1768 of OVOL1 3′ UTRhsa-miR-30c-5p	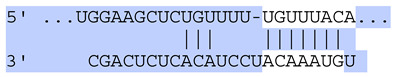	8mer	−0.36
Position 1761–1768 of OVOL1 3′ UTRhsa-miR-30a-5p	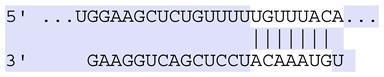	8mer	−0.34
Position 505–511 of OVOL1 3′ UTRhsa-miR-125b-5p	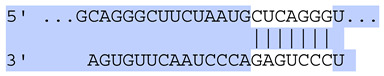	7mer-m8	−0.21
Position 132–138 of NFKBIZ 3′ UTRhsa-miR-374a-5p	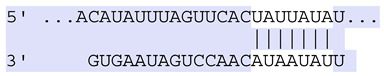	7mer-m8	−0.02
Position 188–194 of FGF10 3′ UTRhsa-miR-425-5p	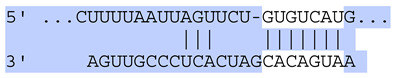	7mer-m8	−0.19
Position 832–838 of FGF10 3′ UTRhsa-miR-19a-3p	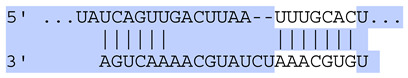	7mer-m8	−0.22
Position 832–838 of FGF10 3′ UTRhsa-miR-19b-3p	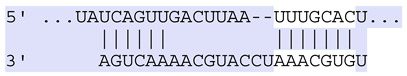	7mer-m8	−0.22
Position 457–463 of FGF10 3′ UTRhsa-miR-302b-3p	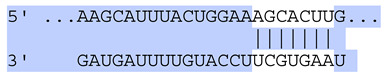	7mer-m8	−0.21
Position 483–489 of FGF10 3′ UTRhsa-miR-9-5p	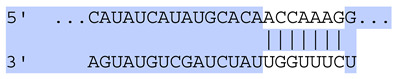	7mer-m8	−0.16
Position 3632–3638 of EREG 3′ UTRhsa-miR-19b-3p	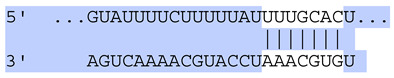	7mer-m8	−0.32
Position 3632–3638 of EREG 3′ UTRhsa-miR-19a-3p	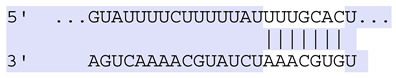	7mer-m8	−0.32
Position 3648–3654 of EREG 3′ UTRhsa-miR-93-5p	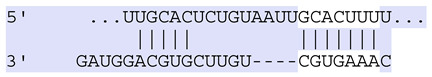	7mer-m8	−0.31
Position 868–874 of PTCH1 3′ UTRhsa-miR-15b-5p	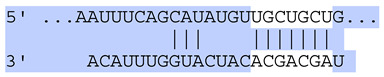	7mer-m8	−0.20
Position 868–874 of PTCH1 3′ UTRhsa-miR-15a-5p	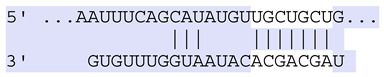	7mer-m8	−0.20
Position 2854–2860 of PTCH1 3′ UTRhsa-miR-200c-3p	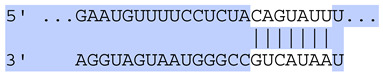	7mer-m8	−0.10
Position 547–553 of PTCH1 3′ UTRhsa-miR-9-5p	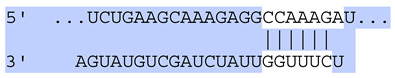	7mer-A1	−0.05
Position 83–89 of CDH13 3′ UTRhsa-miR-30e-5p	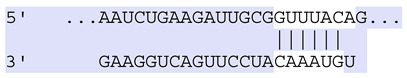	7mer-A1	−0.17
Position 83–89 of CDH13 3′ UTRhsa-miR-30c-5p	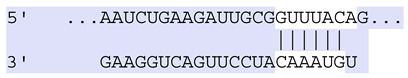	7mer-A1	−0.17
Position 37–43 of SDR16C5 3′ UTRhsa-miR-194-5p	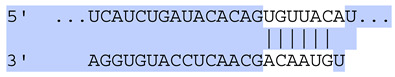	7mer-A1	−0.13
Position 263–269 of IVL 3′ UTRhsa-miR-200c-3p	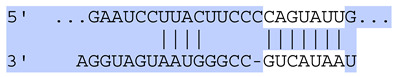	7mer-m8	−0.13
Position 222–228 of IVL 3′ UTRhsa-miR-26a-5p	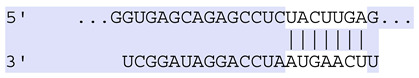	7mer-m8	−0.02
Position 264–270 of KRT10 3′ UTRhsa-miR-93-5p	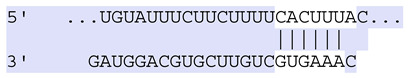	7mer-A1	−0.09
Position 281–287 of KRT10 3′ UTRhsa-miR-186-5p	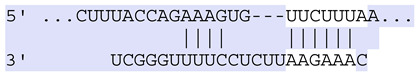	7mer-A1	−0.06
Position 1327–1333 of CDH1 3′ UTRhsa-miR-9-5p	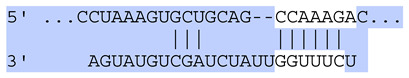	7mer-A1	−0.06
Position 70–76 of DSG1 3′ UTRhsa-miR-23a-3p	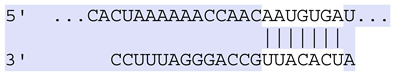	7mer-m8	−0.10
Position 83–90 of DSG1 3′ UTRhsa-miR-210-3p	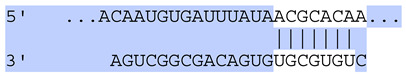	8mer	−0.54
Position 369–375 of DSG1 3′ UTRhsa-miR-32-5p	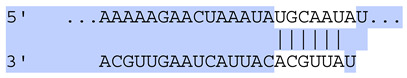	7mer-A1	−0.11
Position 369–375 of DSG1 3′ UTRhsa-miR-92a-3p	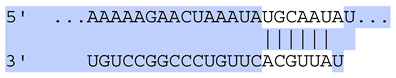	7mer-A1	−0.08
Position 651–657 of DSG1 3′ UTRhsa-miR-24-3p	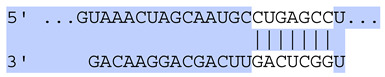	7mer-m8	−0.05
Position 2044–2050 of DSG1 3′ UTRhsa-miR-26a-5p	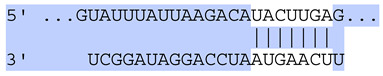	7mer-m8	−0.07
Position 883–890 of ITGA3 3′ UTRhsa-miR-181a-5p	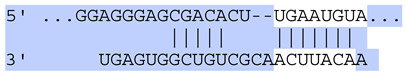	8mer	−0.03
Position 883–890 of ITGA3 3′ UTRhsa-miR-181b-5p	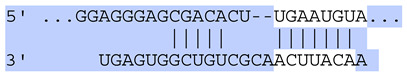	8mer	−0.03
Position 630–637 of CERS3 3′ UTRhsa-miR-150-5p	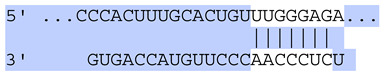	8mer	−0.11
Position 1157–1163 of CYP26B1 3′ UTRhsa-miR-100-5p	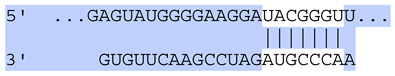	7mer-m8	−0.25
Position 2804–2810 of CYP26B1 3′ UTRhsa-miR-302c-3p	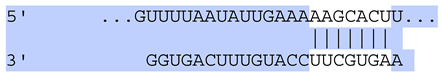	7mer-m8	−0.28
Position 36–42 of KLK5 3′ UTRhsa-miR-103a-3p	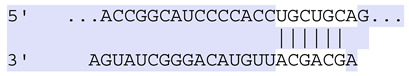	7mer-A1	−0.13
Position 55–61 of KLK5 3′ UTRhsa-miR-122-5p	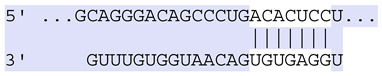	7mer-m8	−0.23
Position 101–107 of KLK5 3′ UTRhsa-miR-143-3p	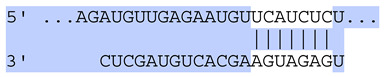	7mer-m8	−0.23
Position 81–87 of FLG 3′ UTRhsa-miR-125b-5p	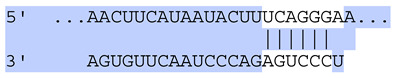	7mer-A1	−0.16

## Data Availability

The original contributions presented in the study are included in the article and Appendix A, further inquiries can be directed to the corresponding author.

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
