# Peer review of "Novel Exosomal miRNA Expression in Irradiated Human Keratinocytes"

_ijms, 2024, doi:10.3390/ijms252212477_

Round 1
Reviewer 1 Report
Comments and Suggestions for Authors
This study aimed to investigate the impact of Sun2000 solar simulator irradiation on exosomal miRNA profiles in human keratinocytes. The results indicated that the size of exosome increased in irradiated cells. Fifty-nine exosomal miRNAs were differentially expressed in irradiated human keratinocytes, including hsa-miR-25 425-5p, hsa-miR-181b-5p, hsa-miR-196b-5p, hsa-miR-376c-3p, and hsa-miR-15a-5p.
1. The cell death and ROS generation caused by UV irradiation was well-known and many studies demonstrated this effect. The changes of exosomal miRNAs in human keratinocytes after UV exposure were present in this study. However, the relationship of UV irradiation, ROS generation and the expression of exosomal miRNAs were unclearly in this study. In addition, the novelty of this study must describe clearly in the manuscript.
2. This manuscript is interest and many information are present. Authors stated that “This study high-lights the significant impact of solar radiation on exosomal miRNA expression in keratinocytes, suggesting their potential role in the cellular response to oxidative stress.” However, this study only showed the expression of the gene after UV-irradiation rather than the activities or functions of these gene as well as this gene to ROS or UV irradiation. The information provided by this study was limited.
3. Figure 2 showed the true relative ratio of fluorescence intensity of human keratinocytes. immediately and 24 hr post solar simulator irradiation at different exposure times. This Figure was confused, and must explain clearly in the Figure Legends and text.
4. The results of this study have to compared and discussed with other studies for clearly describing the findings of this study.
5. The abbreviation must define at the first time shown in the text.
Comments on the Quality of English LanguageThe English could be improved to more clearly express the research.
Author Response
Reviewer 1:
This study aimed to investigate the impact of Sun2000 solar simulator irradiation on exosomal miRNA profiles in human keratinocytes. The results indicated that the size of exosome increased in irradiated cells. Fifty-nine exosomal miRNAs were differentially expressed in irradiated human keratinocytes, including hsa-miR-25 425-5p, hsa-miR-181b-5p, hsa-miR-196b-5p, hsa-miR-376c-3p, and hsa-miR-15a-5p.
We thank the reviewer for the valuable feedback and for taking the time to help this work improve; we hereby address and incorporate suggestions and changes to the manuscript for all of the issues they raised;
- The cell death and ROS generation caused by UV irradiation was well-known and many studies demonstrated this effect. The changes of exosomal miRNAs in human keratinocytes after UV exposure were present in this study. However, the relationship of UV irradiation, ROS generation and the expression of exosomal miRNAs were unclearly in this study. In addition, the novelty of this study must describe clearly in the manuscript.
We thank the reviewer for pointing this out. We have revised the manuscript and we have added a description to clarify the relationship of UV irradiation, ROS generation and the expression of exosomal miRNAs ). In addition, we added a statement to justify the novelty of this study (Line 140)
- This manuscript is interest and many information are present. Authors stated that “This study high-lights the significant impact of solar radiation on exosomal miRNA expression in keratinocytes, suggesting their potential role in the cellular response to oxidative stress.” However, this study only showed the expression of the gene after UV-irradiation rather than the activities or functions of these gene as well as this gene to ROS or UV irradiation. The information provided by this study was limited.
This observation is correct. To provide a clearer picture of the relationship, we provide a comprehensive experimental design that considers the hermetic effect. This describes the time-dependent adaptive response of mammalian cells to oxidative stress. This phenomenon plays a crucial role in adapting to various stressors, including oxidative stress, by upregulating antioxidant enzymes and protective mechanisms, allowing cells to adapt and survive the stress for a defined period, typically lasting 18-24 hours. Based on this, HEKa cells were exposed to varying durations of solar simulator irradiation (10,20,30,40,50, and 60 minutes). To evaluate the presence of acute oxidative stress, intracellular ROS levels were quantified using the CellROX Deep Red Assay immediately after irradiation for each exposure time point. A return of ROS levels to baseline within a 24-hour timeframe was indicative of an acute oxidative stress response. Fold change analysis was used to identify potential HEKa adaptation to the acute oxidative stress response after solar simulator irradiation. Significant reduction in fold change indicating the onset of an adaptive response characterized by an initial increase in ROS after irradiation, followed by a return to baseline levels within 24 hours. The most significant reduction in ROS levels (fold change of -0.747) occurred at the 40-minute exposure. Alongside cell viability was assessed using the colorimetric MTT assay following exposure to solar simulator for varying durations (10, 20, 30, 40, 50, and 60 minutes). Three independent biological replicates were performed, with at least three technical replicates within each biological replicate to enhance data reliability. This was added to the manuscript (Line 186/ highlighted). Moreover, we add the functional analysis on miRNA targets to identify and validate the target genes of the differentially expressed miRNAs hoping this would provide insights into their potential roles in cellular responses to oxidative stress using Target Scan Human as a bioinformatic database (Line 359). Given the critical roles of proliferation, adhesion, terminal differentiation, and cornification in maintaining the cellular homeostasis of keratinocytes under oxidative stress, we focused on identifying target genes associated with these processes. In future work we plan to integrate a dual- luciferase reporter assay (described Line 490) to further clarify the validation of predicted target genes and their corresponding exosomal miRNA binding sites.
- Figure 2 showed the true relative ratio of fluorescence intensity of human keratinocytes. immediately and 24 hr post solar simulator irradiation at different exposure times. This Figure was confused, and must explain clearly in the Figure Legends and text.
This was an oversight. We have added an illustration to show what’s been included, and how the difference between control cells and irradiated cells was calculated. We have adjusted the image so that it is clearer and more readable. Figure legend is also adjusted to meet the above comment.
- The results of this study have to compared and discussed with other studies for clearly describing the findings of this study.
We agree, accordingly, we have revised the discussion section and extended our manuscript to compare our results to other studies in the field (Line 389).
- The abbreviation must define at the first time shown in the text.
We agree with this and have incorporated the abbreviations definitions at first times throughout the manuscript.
Reviewer 2 Report
Comments and Suggestions for Authors
The manuscript titled “Unique Exosomal miRNA Expression in Irradiated Human Keratinocytes” is the result of a considerable piece of scientific work. Although the manuscript presents some potential it presents also several issues that need to be addressed.
Line 2 Please consider removing the “Unique” form your title since you have investigated all the miRNAs upon irradiation and not only the “unique” ones.
Lines 137 – 151 The last paragraph of the introduction must clearly state only study’s purpose and justification. The results and conclusion are not reported here. It is suggested that these lines are rewritten in a separate paragraph.
Lines 153 – 164 belong to the ‘materials and methods” part.
Lines 215 – 219 belong to the ‘materials and methods” part.
Lines 220 – 223 you provide a regression analysis result (probably linear) but in the materials and methods part you do not state any regression analysis, or the software used for this purpose (is it still Excel?). Moreover, you say “revealed a normal distribution based on the Q-Q plot” but when speaking for normality of distributions is highly recommended to undergo the Shapiro-wilk test to establish the fulfillment of such criterion. Last if you skeak for a Q-Q plot it is recommended to provide it.
Lines 228 – 231 belong to the materials and methods part.
Lines 226 – 256 The part needs an extensive rewrite. The statistical analysis between irradiation regimens must be reported as along as the statistical analysis within groups. Especially the lines 253 – 255 are not understandable.
Lines 291 – 336 This part is also not understandable. What is being compared to what. The within irradiation regimes variations? Different irradiation regimes? Each regimen with the control?
The discussion part is also narrow and cannot be reviewed without knowledge of the results.
Lines 405 – 406 the part “from an ABET Technology Sun2000 solar simulator” must be erased since it is reported in the previous sentence
Line 406 Was it triplicate for every time exposure regimen?
Line 407 Please erase “ABET Technology”. Already reported
Lines 401 – 415 Since I am not familiar with Sun2000 I do not know if the distance between the source and the specimen irradiated is standard or adjustable. Please explain if the source is standard the distance, and if adjustable the distance between the source and the specimen.
Line 459 Nunc is a brand name thus the company must be stated in parenthesis
Line 460 As technical replicate, do you refer to the preparations for measurement? Meaning that you took from each biological sample three samples and measured them? If it is the case, wouldn’t it be better to state that every measurement was performed in triplicate?
Lines 459 – 467 According to my understanding the procedure for exosome isolation was performed to the supernatant. But how did you establish that those cultures had the same cell populations? It is well established that the quantity of UV irradiation is related to cell viability. If you want to have comparative results between cells of different cultures, for procedures that implement viable cells, you must have the same conditions (not the same as in oxstress). In this case it seems that we will not really know whether the number of exosomes corresponds to a reduced or enhanced cell population and thus the results seem not to be comparable between different solar irradiation dose regimens.
Line 466 “ThedicownhndNe” please specify
Line 499 Again the quantification should be referred to an initial cell population to be comparable between two different irradiation regiments
Line 508 Quantitative analysis here has the same problem as previously
Line 532 The statistical analysis with parametric methods must follow the check for normality of distribution (Shapiro - wilk). If the data does not exhibit normal distribution, then nonparametric test shall be used.
Comments on the Quality of English LanguageThe manuscript need moderate language improvement.
Author Response
The manuscript titled “Unique Exosomal miRNA Expression in Irradiated Human Keratinocytes” is the result of a considerable piece of scientific work. Although the manuscript presents some potential it presents also several issues that need to be addressed.
We are grateful to the reviewers for their insightful comments on this manuscript. We have been able to incorporate changes to reflect most of the suggestions provided by the reviewer. We have highlighted the changes within the manuscript.
Line 2 Please consider removing the “Unique” form your title since you have investigated all the miRNAs upon irradiation and not only the “unique” ones.
Thank you for pointing this out. We agree with this comment that it might create a confusion as we have investigated all changes not just unique miRNAs, we have replaced the word with “Novel”.
Lines 137 – 151 The last paragraph of the introduction must clearly state only study’s purpose and justification. The results and conclusion are not reported here. It is suggested that these lines are rewritten in a separate paragraph.
We agree, accordingly, we have revised the paragraph so that it only states the study purpose and justification Line 140.
Lines 153 – 164 belong to the ‘materials and methods” part.
We agree, text moved to line 578 (materials and methods)
Lines 215 – 219 belong to the ‘materials and methods” part.
We understand there are some technical details related to methods that are added in the section mentioned, we decided to put it in results as it includes results in the text and to provide clarification to the workflow. This is the text (Lines 220 – 224) with the result (highlighted):
“Fold change analysis was used to identify potential HEKa adaptation to the acute oxidative stress response after solar simulator irradiation. Significant reduction in fold change indicating the onset of an adaptive response characterized by an initial increase in ROS after irradiation, followed by a return to baseline levels within 24 hours. The most significant reduction in ROS levels (fold change of -0.747) occurred at the 40-minute exposure, (as shown in Table 1).”
Therefore, we think it will be clearer if we keep this section in the manuscript.
Lines 220 – 223 you provide a regression analysis result (probably linear) but in the materials and methods part you do not state any regression analysis, or the software used for this purpose (is it still Excel?). Moreover, you say “revealed a normal distribution based on the Q-Q plot” but when speaking for normality of distributions is highly recommended to undergo the Shapiro-wilk test to establish the fulfillment of such criterion. Last if you skeak for a Q-Q plot it is recommended to provide it.
We thank the author for this comment, the software name is JMP16 software. We have done a Shapiro-wilk test and it confirmed a normal distribution, We completely agree about providing details about the statistical analysis, therefore we incorporated the details in the (Lines: 183, 727). Also the analysis is provided as supplementary data (Supplementary Figure 1 (a,b)).
Lines 228 – 231 belong to the materials and methods part.
We agree, text moved to line 600 (materials and methods)
Lines 226 – 256 The part needs an extensive rewrite. The statistical analysis between irradiation regimens must be reported as along as the statistical analysis within groups. Especially the lines 253 – 255 are not understandable.
The paragraph was rewritten taking into consideration the mentioned above, Line 235.
Lines 291 – 336 This part is also not understandable. What is being compared to what. The within irradiation regimes variations? Different irradiation regimes? Each regimen with the control?
We agree this was not clear, we adjusted the paragraph and clarified as highlighted Line 313, the exosomal miRNA analysis compared the irradiated HEKa cells to the non-irradiated as explained in the paragraph after rewriting
The discussion part is also narrow and cannot be reviewed without knowledge of the results.
We agree, accordingly, we have revised the discussion section and extended our manuscript to compare our results to other studies in the field (Line 389).
Lines 405 – 406 the part “from an ABET Technology Sun2000 solar simulator” must be erased since it is reported in the previous sentence
Thank you for pointing this out, we have fixed the error.
Line 406 Was it triplicate for every time exposure regimen?
Yes they were, this is added to materials and methods (Line 547)
Line 407 Please erase “ABET Technology”. Already reported
Thank you for pointing this out, we have fixed the error.
Lines 401 – 415 Since I am not familiar with Sun2000 I do not know if the distance between the source and the specimen irradiated is standard or adjustable. Please explain if the source is standard the distance, and if adjustable the distance between the source and the specimen.
We thank you for providing your expertise in this matter. Yes, the distance is standard, this is added to materials and methods line 543
Line 459 Nunc is a brand name thus the company must be stated in parenthesis
Thank you for spotting that. Company name is added to line 611
Line 460 As technical replicate, do you refer to the preparations for measurement? Meaning that you took from each biological sample three samples and measured them? If it is the case, wouldn’t it be better to state that every measurement was performed in triplicate?
Thank you for pointing that out, a statement that every measurement was performed in triplicates is added to line 611
Lines 459 – 467 According to my understanding the procedure for exosome isolation was performed to the supernatant. But how did you establish that those cultures had the same cell populations? It is well established that the quantity of UV irradiation is related to cell viability. If you want to have comparative results between cells of different cultures, for procedures that implement viable cells, you must have the same conditions (not the same as in oxstress). In this case it seems that we will not really know whether the number of exosomes corresponds to a reduced or enhanced cell population and thus the results seem not to be comparable between different solar irradiation dose regimens.
This is a very good point, we have calculated cell number using an automated cell counter (these details are added to materials and methods Line 533), this was done before any experiment and all cells were seeded at the optimal density. Experiments were performed when cell reached 60% confluency ( this yields 6 million cells for 75 cm flasks). For exosome and miRNA analysis only one irradiation regimen was used (40 minutes irradiation). At 40 minutes exposure, our data showed a decrease in cell viability up to 78% immediately after irradiation, the supernatant was collected immediately, and we looked at the size and miRNA content differences of the exosomes between the groups and not the differences in the numbers of exosomes.
Line 466 “ThedicownhndNe” please specify
We apologise for this typological mistake; the sentence is corrected (Line 618)
Line 499 Again the quantification should be referred to an initial cell population to be comparable between two different irradiation regiments
We agree, in the cell culture protocol all cells were seeded at the same density 2500 cells/cm² prior to any experiment. The only difference between irradiated and control is the solar simulator irradiation. For miRNA analysis, the dose used for irradiation was 40 minutes, as this is the dose that caused the most acute response as explained in section 2.3/2.4 and their effect on cell viability was not statistically significantly different from 20 minutes and 30 minutes. However, at 50 minutes the cell viability dropped to half. Moreover, 40 minutes exposure had the highest fold change when compared to 24 hours post irradiation, which indicated the acute effect on cell viability, this is clarified and added to section 2.3 line 226
Line 508 Quantitative analysis here has the same problem as previously
Adjusted to explain seeding density for both control and irradiated cells, we understand this should have been explained before so we appreciate the valuable feedback; we included a clarification.
Line 532 The statistical analysis with parametric methods must follow the check for normality of distribution (Shapiro - wilk). If the data does not exhibit normal distribution, then nonparametric test shall be used.
Addressed and adjusted as explained above.
Round 2
Reviewer 2 Report
Comments and Suggestions for Authors
The authors addressed all mz comments, and significantly improved their manuscript. In my view, this article meets now the quality standards of the present journal, and I therefore recommend it for publication.
Author Response
Comment 1: "The authors addressed all mz comments, and significantly improved their manuscript. In my view, this article meets now the quality standards of the present journal, and I therefore recommend it for publication."
Reply: We sincerely appreciate the time and effort that you and the reviewers dedicated to providing feedback on our manuscript. We are grateful for the insightful comments and the improvements to the paper.
